# Expressed Soybean Leghemoglobin: Effect on *Escherichia coli* at Oxidative and Nitrosative Stress

**DOI:** 10.3390/molecules26237207

**Published:** 2021-11-28

**Authors:** Olga V. Kosmachevskaya, Elvira I. Nasybullina, Konstantin B. Shumaev, Alexey F. Topunov

**Affiliations:** Bach Institute of Biochemistry, Research Center of Biotechnology of the Russian Academy of Sciences, 119071 Moscow, Russia; rizobium@yandex.ru (O.V.K.); lvirus198709@rambler.ru (E.I.N.); tomorov@mail.ru (K.B.S.)

**Keywords:** leghemoglobin, *Escherichia coli*, protein expression, dinitrosyl iron complexes, spectrophotometry, electron paramagnetic resonance, oxidative stress, nitrosative stress

## Abstract

Leghemoglobin (Lb) is an oxygen-binding plant hemoglobin of legume nodules, which participates in the symbiotic nitrogen fixation process. Another way to obtain Lb is its expression in bacteria, yeasts, or other organisms. This is promising for both obtaining Lb in the necessary quantity and scrutinizing it in model systems, e.g., its interaction with reactive oxygen (ROS) and nitrogen (RNS) species. The main goal of the work was to study how Lb expression affected the ability of *Escherichia coli* cells to tolerate oxidative and nitrosative stress. The bacterium *E. coli* with the embedded gene of soybean leghemoglobin *a* contains this protein in an active oxygenated state. The interaction of the expressed Lb with oxidative and nitrosative stress inducers (nitrosoglutathione, *tert*-butyl hydroperoxide, and benzylviologen) was studied by enzymatic methods and spectrophotometry. Lb formed NO complexes with heme-nitrosylLb or nonheme iron-dinitrosyl iron complexes (DNICs). The formation of Lb-bound DNICs was also detected by low-temperature electron paramagnetic resonance spectroscopy. Lb displayed peroxidase activity and catalyzed the reduction of organic peroxides. Despite this, *E. coli*-synthesized Lb were more sensitive to stress inducers. This might be due to the energy demand required by the Lb synthesis, as an alien protein consumes bacterial resources and thereby decreases adaptive potential of *E. coli*.

## 1. Introduction

Leghemoglobin (Lb) is a symbiotic plant hemoglobin (Hb) synthesized in legume nodules that are formed on roots following their infection by nitrogen-fixing nodulating bacteria of several genera (*Rhizobium*, *Bradyrhizobium*, *Azorhisobium*, etc.). In nodules, these bacteria change their form and properties, turning into a symbiotic state bacteroids. Lb is necessary for a nitrogen fixation by the symbiotic system of legume plants. The oxygen (O_2_) concentration in the nodule central zone is very low and nontoxic for bacterial nitrogenase, the enzyme reducing molecular nitrogen to ammonia [1,2,3]. Due to its remarkable affinity to oxygen (p_50_ ~ 40–70 nM) [4,5,6], Lb maintains oxygen flow to fast-breathing bacteroids in the central nodule zone [7,8,9]. The RNA sequencing analysis shows that Lb is expressed within a few hours after the inoculation and that its mRNA is also found in roots and pods [10]. This indicates that Lb functions are not confined to nitrogen fixation only.

Lb is found in the cytoplasm of infected plant nodule cells, with the content reaching 30–40% of the total soluble proteins (up to a millimolar concentration) [11,12]. This amount is so high that it is extremely difficult to model it in vitro in an artificial system containing Lb and bacteroids. Lb function in plant cells is similar to myoglobin (Mb) in the muscle tissue cells. Lb, like Mb, is an O_2_ depot, a constant supply of oxygen, independent of its input/output [13]. These hemoproteins are not only similar in functions but also in tertiary structures, up to the location of α-helixes [14]. Plant and animal hemoglobins had a common ancestral gene, while the plant Hb gene lost one intron during evolution [3], which led to a difference in the amino acid sequences of these proteins.

In addition to oxygen binding, Lb performs a number of other functions, which may be useful under oxidative and nitrosative stress. It was suggested that Lb regulates the nitric oxide (NO) concentration [15]. Lb can also function as a peroxidase-reducing organic peroxides [16] and catalyze the isomerization of peroxynitrite into nitrate [15,17]. Probably, due to these reactions, Lb halts some unregulated plant protective reactions, triggered by reactive nitrogen (RNS) and reactive oxygen species (ROS) during the formation of nitrogen-fixing symbiosis [18]. In plants, Lb activities such as peroxidase, NO-dioxygenase, and peroxynitrite isomerase can play a pivotal role in the overall antioxidant protection of the infected legume nodule cell, when Lb concentration is very high.

Meanwhile, the interaction of Lb with lipid hydroperoxides and hydrogen peroxide leads to the formation of the oxoferryl Lb form (Heme-Fe^IV^=O), a strong oxidizer, that triggers lipid peroxidations reactions (POL) in cell membranes [19]. Moreover, during these processes, an oxidative modification of Lb happens, leading to the degradation of heme and the release of iron ions. The latter catalyzes free radical formation in the Fenton and Haber–Weiss reactions [20].

Under the same conditions, the rate of oxoferryl Lb formation is 5 times higher than that of Mb transformation to oxoferryl intermediate [16]. The link of the heme group with the protein in Lb is weaker than in Mb [21]. In case of reacting with peroxides, Lb heme is destroyed, releasing iron, which, in turn, catalyzes the formation of hydroxyl radical. In this case, Lb heme is destroyed faster than the Mb one. In addition, Lb is much more prone to auto-oxidation (Lb^II^O_2_ → Lb^III^ + O_2_^•^) than Mb [22]. All these facts indicate that the pro-oxidant potential of Lb is higher than of Mb.

For a long time, Lb was investigated only as a component of a nitrogen-fixing symbiotic system, which determines the efficiency of legume-rhizobial symbiosis and, finally, the yield of legumes. Recent years have witnessed a revival of interest to this protein. In the food industry, there is a growing demand for products from vegetable raw materials that would successfully imitate meat products of animal origin (so-called plant-based meat). It turned out that Lb is a plant hemoprotein that can serve as an ingredient for substituting the meat-like organoleptic properties to a plant base [23,24]. In this regard, there is a need to produce Lb on an industrial scale. The use of soybean Lb on this scale is the most promising, as it shows the highest stability under the impact of different environmental factors as compared with other leghemoglobins [25].

Since Lb concentration in the cytosol of the nodule cells reaches several millimoles, nodules are a convenient source of Lb for studies. An alternative way to obtain Lb is its expression in bacteria, yeasts, and other eukaryotic organisms. These expressing systems can be used both for obtaining Lb and for studying it at various experimental conditions.

The first Lb expression was performed in 1995 in our collaboration with the Institute of Bioorganic Chemistry of the Polish Academy of Sciences (Poznan, Poland) and the Laboratory of Molecular Biology (Hamburg, Germany) [26]. It was the expression of lupin LbI in *E. coli* (strain BL21). Later, soybean Lb*a* [22,27], cowpea LbII [28], and nonsymbiotic plant Hb (rice Hb1) [29] were expressed in *E. coli*. Lb in expressing cells was in a physiologically active reduced state and, in all cases, similar to the nodule protein.

In our work on the expression of lupin LbI [26], we suggested that bacterial reductases could contribute to maintaining the expressed Lb in a reduced state. We demonstrated the presence of reductases in bacteria, which reduce oxygen-carrying hemoproteins [30]. Subsequently, the enzyme-reducing Lb was isolated from *Bradyrhizobium lupini* bacteria and bacteroids [31]. The Kundu group isolated methemoglobin reductases from cyanobacterium *Synechocystis* sp. [32] and green alga *Chlamydomonas reinhardtii* [33]. All these reductases were FAD-containing NADH-dependent enzymes and similar to metLb reductase (ferricLb reductase), reducing Lb in nodules [34,35]. The soybean ferricLb reductase was expressed in *E. coli* itself [36]. As for nonsymbiotic plant hemoglobins, Igamberdiev et al. showed the reduction of barley Hb1 by monodehydroascorbate reductase that reduced Hb1 via ascorbate [37,38]. NAD(P)H, ascorbate, reduced glutathione, and cysteine can be potential nonenzymatic reducers of Lb^III^ to Lb^II^ [39].

At the same time, Lb is an alien protein for *E. coli* cells—the synthesis of which consumes a significant share of biochemical and energy resources, which can affect their stability.

Taking all this into account, we studied how Lb expression affected the ability of *E. coli* cells to tolerate oxidative and nitrosative stresses.

## 2. Results

### 2.1. The Effect of Different Concentrations of Inducers of Oxidative and Nitrosative Stress Inductors on the Growth of E. coli Cells

In this work, we used three types of cell cultures: nonsynthesizing Lb (C_Lb−_)—0.08 µg of heme/mg protein; synthesizing Lb*a* in small amounts (C_Lb+_)—0.15 µg of heme/mg protein; and actively synthesizing Lb*a* (C_Lb++_)—0.45 µg of heme/mg protein.

The *E. coli* strain TB-1 (pEMBL18+: SyLba) used in our study had a high level of Lb expression, which was evidenced by the pronounced pink color of the cells grown at the liquid nutrient medium LB. The large amount of Lb*a* allows spectrophotometrically registering Lb state in whole cells (Figure 1). Their absorption spectra revealed some peaks typical of oxygenated Lb: 411 nm (the Soret region), 541, and 575 nm [40] This indicates that unlike many other *E. coli* expression systems, the system used in our work allowed to obtain Lb in a soluble physiologically active state, identical to the soybean Lb from nodules.

The following substances were used to induce oxidative and nitrosative stress conditions in *E. coli* cells: S-nitrosoglutathione (GSNO), *tert*-butyl hydroperoxide (*t*-BOOH), and benzylviologene (Bv). GSNO is one of the main metabolites in living systems. GSNO is a physiological NO metabolite in the organism and, at the same time, NO donor in living systems. *t*-BOOH is an analog of organic hydroperoxides formed under the conditions of oxidative stress at the process of free radical peroxidation. Bv, like *t*-BOOH, is widely used for modeling of oxidative stress. Since Bv has the ability of single-electron oxidation and reduction, its biological activity is associated with electron transfer from the components of respiratory chain to oxygen with the formation of superoxide (O_2_^•−^), which is a precursor of other reactive oxygen (ROS) and reactive nitrogen species (RNS), including hydrogen peroxide and peroxynitrite [41]. The structural formulas of GSNO, *t*-BOOH, and Bv are shown on Figure 2d.

First, we assessed the effect of different concentrations of oxidative and nitrosative stress inducers on the growth of *E. coli* cells. It was shown that all examined substances inhibited the cell growth, and the inhibition concentration curve was obtained (Figure 2). GSNO at the concentrations up to 0.2 mM did not inhibit the growth of bacterial cultures. The most powerful inducer of oxidative stress was benzylviologen. Starting from a concentration of 0.25 mM, it suppressed cell growth completely. For further experiments, the concentration of GSNO (0.2 mM), which had a slight effect on cell growth, the concentration of *t*-BOOH (0.04 mM), and Bv (0.125 mM), which allowed cells to subsequently restore their ability to grow, were selected.

### 2.2. The Effect of Lb Expression on the Growth of E. coli Cells

In our experiments, the growth of all cultures in the middle of the exponential phase (7 h) was suppressed by *t*-BOOH (Figure 3). *E. coli* cells expressing Lb at a low level (C_Lb+_) had greater sensitivity to *t*-BOOH: the level of their growth by 7 h compared to the control was 43%, while C_Lb−_ and C_Lb++_ ~ 65 %.

It should be noted, that the minimum on the growth curves of Lb synthesizing cells was observed in the exponential phase (7–9 h) under the action of *t*-BOOH only or *t*-BOOH together with GSNO (Figure 3b,d). Moreover, in case of C_Lb+_, it was much more noticeable than in C_Lb++_. This may arise from the oxoferryl Lb form (Heme-Fe^IV^=O^•^) formed under the action of organic peroxides [19]. The following reactions (1)–(4) can lead to the formation of ferrylLb, alkoxy- (RO^•^), and alkyl peroxyl (ROO^•^) radicals:Lb-Fe^III^ + ROOH → Lb-Fe^IV^=O + RO^•^ + H^+^(1)
Lb-Fe^III^ + ROOH → Lb^•^-Fe^IV^=O + ROH(2)
Lb^•^-Fe^IV^=O + ROOH → Lb-Fe^IV^=O + ROO^•^ + H^+^(3)
Lb-Fe^IV^=O + ROOH → Lb-Fe^III^-OH + ROO^•^(4)

In an H_2_O_2_ reaction with Lb, the destruction of the heme group can also proceed with the release of iron, which catalyzes the formation of a highly reactive hydroxyl radical (^•^OH) in the Fenton reaction. OxyLb is especially sensitive to the peroxides’ action [20].

The growth of all types of *E. coli* cells under the conditions of *t*-BOOH-induced oxidative stress was stimulated by GSNO (Figure 3b,d). The protective effect of GSNO was observed even at *t*-BOOH concentration, which caused the complete suppression of the growth of studied strains.

The effect of GSNO on bacterial cells can be explained by the reactions of NO with oxoferryl hemoproteins [42] (reaction (5)) and organic radicals (reactions (6)–(8)) [43], both arising during oxidative stress:Heme-Fe^IV^=O^•^ + NO^•^ → Heme-Fe^III^-ONO → Heme-Fe^III^ + NO_2_^−^(5)
ROO^•^ + NO^•^ → ROONO → RO^•^ + NO_2_^•^(6)
RO^•^ + NO^•^ → RNO_2_(7)
RO^•^+ NO^•^ → RONO_2_(8)

In these reactions, organic nitrates (RNO_2_, RONO_2_) and organic nitrosoperoxy complexes (ROONO) are formed, which, unlike peroxynitrite, do not have strong pro-oxidant properties [43].

Nitric oxide is also able to react with a hydroxyl radical:NO^•^ + ^•^OH → HNO_2_ ↔ NO_2_^−^ + H(9)

Heme-Fe^II^-NO can reduce *t*-BOOH according to the reaction [44]:Heme-Fe^II^-NO + *t*-BOOH +H_2_O → Heme-Fe^III^ + *t*-BOH + HNO_2_ + OH^−^(10)

At the same time, the effect of the combined action of GSNO and Bv significantly differed from that of the GSNO action together with *t*-BOOH (Figure 3d,e). In case of the combined action of GSNO and Bv, the protective effect of GSNO was not observed (Figure 3c,e). The combined action of Bv and GSNO may produce peroxynitrite (ONOO^−^), which is a much stronger oxidizer than O_2_^•−^. Peroxynitrite can be also formed in the NO reaction with O_2_^•−^, released during the auto-oxidation of oxyHb. Thus, oxidative stress induced by *t*-BOOH is the conditions, when NO metabolites can act as the first line of bacterial cell defense.

### 2.3. Effect of Lb Expression on Heme Concentration and Peroxidase Activity in Cellular Protein Extracts

Since the synthesis of antioxidant enzymes is stress induced, we decided to test which way oxidative and nitrosative stress affect the peroxidase activity of *E. coli* cells. For this purpose, the peroxidase activity was measured in crude protein extracts, with *o*-dianisidine as a reducing substrate. The peroxidase activity differed greatly depending on the cell type (Figure 4b). In the case of the actively synthesized Lb culture, the peroxidase activity in all variants of the experiment was almost an order higher than that of C_Lb−_ and C_Lb+_: in C_Lb++_—1.15 nkat/mg protein (control cells without additives) and in C_Lb−_—0.05 nkat/mg protein.

The effect of oxidative and nitrosative stress inducers on peroxidase activity was also dependent on the type of culture (Figure 4b). *t*-BOOH, Bv, and GSNO decreased the peroxidase activity in the C_Lb−_ culture by 2–3 times. To a greater extent, the decrease occurred in the presence of GSNO (8 times) and GSNO together with Bv (16 times). All inducers stimulated the activity of C_Lb+_ by 20–30%, in contrast to GSNO, which doubled the activity. By contrast, in C_Lb++_, Bv, separately and together with GSNO, suppressed the activity by about 6 times, while it was several times higher than that of C_Lb−_. These changes in peroxidase activity were surprisingly well correlated with the shifts of heme content in the same cell extracts. This is particularly noticeable in C_Lb++_ (Figure 4a). Therefore, under the Bv influence, the oxidative degradation of hemoproteins occurs in *E. coli* cells with a high expression of Lb, which is the major protein here.

Comparing diagrams on Figure 4, we can assume that a significant contribution to the overall peroxidase activity of cell extracts was made by Lb, which catalyzes the reduction of hydrogen peroxide and organic hydroperoxides [16,45].

*E. coli* cells with a low Lb level were the most sensitive to *t*-BOOH and *t*-BOOH together with GSNO action (Figure 3). No matter that C_Lb−_ cells had a minimum impact on the growth curve after adding oxidative and nitrosative stress inducers, they restored growth faster than the Lb-synthesizing cells. It can be concluded that Lb synthesis decreased the adaptive potential of the bacterial cell, although Lb can restore organic peroxides [16,45] and capture peroxynitrite [15,17]. This implies that in the studied system the pro-oxidant Lb effects prevail over the antioxidant ones.

Thus, in all types of the cells examined, the combination of *t*-BOOH and GSNO induces a smaller deviation in the peroxidase activity value if compared to the control. The formation of products not contributing to free radical reactions arising from the interaction of NO with *tert*-butyl radicals (reactions (6)–(8)) may account for this discrepancy. Such an antioxidant effect of NO should lead to a decrease in the synthesis of antioxidant enzymes, including those with peroxidase activity.

### 2.4. Peroxidase Activity of the Isolated Lb

Since Lb is known to form heme complexes with aliphatic acids [46], it may well interact with small lipoperoxides, such as *t*-BOOH. Moreover, Lb is able to catalyze the reduction of peroxides in some cases [16,45]. The peroxidase activity of Lb is about an order higher than that of Mb, but several orders lower than that of true peroxidases [16]. Thus, this Lb activity is often referred to as “pseudoperoxidase”.

Lb interaction with H_2_O_2_ or organic peroxides leads to a duralectra oxidation metLb with the formation of ferryl intermediates (Lb^IV^, Lb^IV^=O_2_) with cation radical of the porphyrin ring, which migrates to the protein with the formation of a tyrosine radical [19,45]:Fe^III^-porphyrin-globin + H_2_O_2_ → Fe^IV^=O-porphyrin^•+^-globin + H_2_O(11)
Fe^IV^=O-porphyrin^•+^-globin-Tyr → Fe^IV^=O-porphyrin-globin-Tyr^•^(12)

In soybean Lb*a*, it is Tyr133, located near the heme [46]. Tyrosine phenoxyl radicals play a key role in the protein inactivation, thus contributing to intermolecular and intramolecular heme-protein crosslinking [19,47,48]. Therefore, the Lb participation in the peroxidase cycle leads to the inactivation of the protein [45].

Soybean Lb*a* reacts with H_2_O_2_ with a rate constant of 5 × 10^3^ M^−1^ s^−1^. The ferryl intermediate is similar to compound II of the true peroxidases and can be an oxidizer in some reactions [47].

We have experimentally shown the ability of Lb*a* to reduce *t*-BOOH. The dependence of the rate of the reaction catalyzed by Lb*a* on *t*-BOOH concentration, in the presence of an artificial reducing substrate *o*-dianisidine, had a classical Michaelis kinetics with V_max_ = 0.18 mM/min and K_m_ = 5.4 ± 0.05 mM (Figure 5). For comparison, the K_m_ values of bacterial catalases/peroxidases for H_2_O_2_ are in the range 3–5 mM at pH 7.0 [49].

Lb can function in peroxidase oxidation/reduction cycle only in the presence of substances reducing oxyferrylLb and ferrylLb to metLb. It has been shown that glutathione (GSH) is an effective low-molecular-weight reducing agent for Lb^IV^=O [50]. In addition, GSH is a hydrophilic molecule and can come into contact with a heme pocket. As a result, an oxidized GSSG and a thiyl radical (GS^•^) are formed according to the reactions:Lb^IV^=O + GSH → Lb^III^ + H_2_O + GS^•^(13)
GS^•^ + GS^•^ → GSSG(14)

Other physiological electron donors directly reducing ferrylLb can be NAD(P)H, flavins, ascorbate, and lipoic acid [51,52,53]. Therefore, the presence of sufficient amounts of physiological reducing agents can prevent both Lb inactivation and POL peroxidation in the presence of H_2_O_2_.

The physiological significance of Lb peroxidase activity is still unknown. However, considering the high concentration of expressed Lb in *E. coli*, it can be assumed that its contribution to antioxidant protection of these cells could be significant at oxidative stress conditions. In this case, the peroxidase activity of Lb can also play an important role in antioxidant protection. The rapid Lb incorporation to peroxidase cycle can be more effective, thus inducing a rapid cellular response to oxidative stress, than de novo synthesis of the cytoplasmic peroxidases. The peroxidase activity of Lb in plants was also assumed to have biological significance during the aging of root nodules [16,19].

We compared the dependence of the initial reaction rates of the peroxidase reaction, catalyzed by Lb*a* and horseradish root peroxidase (HRP), on the concentration of *t*-BOOH (Figure 6). The concentrations of reductants were: 8 µM Lb and 0.8 µM HRP. It can be seen that up to *t*-BOOH concentration 0.65 mM, Lb was comparable in effectiveness to HRP. However, high concentrations of peroxides caused oxidative damage of Lb, and its catalytic function decreased. This distinguishes Lb and other hemoglobins from the genuine peroxidases.

In plants, Lb can still contribute to the elimination of peroxides due to the high content in the infected nodule cells (~0.3 mM, in very effective nodules—2–3 mM) [12]. The peroxidase activity of Lb in plants was also assumed to have biological significance during the aging of root nodules [16,19].

### 2.5. Formation of Nitrosyl Iron Complexes with Leghemoglobin

The antioxidant effect of NO may also be associated with its ability to form heme and nonheme nitrosyl iron complexes, e.g., with hemoproteins. In the first case, heme iron is nitrosylated (Heme-Fe(II)-NO), while in the second case, the dinitrosyl iron complexes (DNICs) associated with cysteine or histidine residues are formed. Both processes were formerly reported for Hb [54,55,56]. We also suggested the possibility of DNICs to interact with Lb [57].

DNICs represents a form of NO depot and transportation in cells and tissues [58,59,60]. In various in vitro and in vivo systems, they manifested antioxidant and antiradical properties [55,56,61], e.g., in reactions with *t*-BOOH derivatives, superoxide (O_2_^•−^), and peroxynitrite. DNICs bound with low-molecular-weight thiol ligands and with Hb protect SH-groups against oxidation [55,56,62], as well as prevent the formation of heme carbonyl derivatives and its degradation [61]. Moreover, DNICs reduce the oxoferryl heme form of Mb [63]. Their protective effects may also be associated with the inclusion of iron released from heme [64]. Another probable mechanism of the DNICs antioxidant effect is reducing *t*-BOOH in a similar way to reaction (10).

To confirm the formation of Lb-bound DNICs, we added DNICs with phosphate ligands, which are easily replaced by protein ones, to the solutions of recombinant Lb*a* and human Hb. Optical absorption spectra of recombinant Lb*a* and its complex with DNICs are shown in Figure 7. Thus, Lb-bound DNICs could be identified by optical spectra using spectrophotometry.

The EPR spectra of DNICs associated with metHb (Hb-DNIC) and metLb (Lb-DNIC) are characterized by such parameters as g-factor: g_1_ = 2.04, g_2_ = 2.03, and g_3_ = 2.014 (Figure 8). These values are determined by ligand nature and by the environment of DNICs’ binding sites.

Figure 8b shows that Lb-DNICs were destroyed in the presence of *t*-BOOH. We have previously shown that Hb-DNICs also decay under the action of *t*-BOOH and H_2_O_2_ [55,56].

The low-temperature EPR spectroscopy was used to register DNIC signals in both control (C_Lb−_) and Lb-synthesizing *E. coli* bacteria (C_Lb++_), when incubated with GSNO or with the latter + *t*-BOOH (Figure 9). The Hb-DNIC and Lb-DNIC EPR spectra were similar to the spectrum of DNICs with low molecular weight thiol ligands [54]. Therefore, we could only estimate the total DNICs level in *E. coli* cells. *t*-BOOH did not induce any significant changes in the EPR spectrum of C_Lb++_ bacterial cells if compared to the control, but when incubated with GSNO, *t*-BOOH decreased the level of DNICs formed in them (Figure 9a,b; spectra 1,2). The results obtained on the C_Lb+_ strain, were almost identical to the control strain C_Lb−_.

The EPR spectrum, appeared in C_Lb++_ cells, was the superposition of two signals: DNIC and Lb^II^-NO (Figure 9b, spectrum 4). The latter was absent in C_Lb−_ cells (Figure 9a, spectrum 3). Apparently, when DNICs are destroyed, NO is released, subsequently leading to nitrosylation of Lb heme iron. Indeed, we have previously found that Hb^II^-NO appears during the destruction of Hb-DNICs under the O_2_^•−^ action [55,56].

The intensity of the DNICs EPR signal in C_Lb−_ bacteria decreased by 35–45% during the incubation with *t*-BOOH. At similar conditions, the DNICs signal in C_Lb++_ culture decreased by 8–12% only. Thus, the oxidative stress induced by *t*-BOOH led to a decrease in the number of DNICs associated with the bacterial cell components. It could be explained by DNICs breakdown during the interaction with free radicals and oxoferryl forms of hemoproteins [63]. A lower DNICs level in C_Lb++_ cells, compared with the C_Lb−_ ones, during the incubation with only GSNO (Figure 9a,b, spectrum 3), may be due to NO elimination during its interaction with oxygenated Lb, i.e., during the dioxygenase reaction:LbO_2_ + NO → Lb^III^ + NO_3_^−^(15)

At the same time, Lb can protect DNICs from oxidative destruction through its own peroxidase activity. This is consistent with the fact that *t*-BOOH causes a more significant decrease in the DNICs level in C_Lb−_ cells (Figure 9a, spectra 3,4) than in C_Lb++_ ones (Figure 9b, spectra 3,4). The possibility of cooperative (Lb plus DNICs) antioxidant action is also confirmed by the fact that the growth of C_Lb++_ culture in the presence of GSNO, and *t*-BOOH was less inhibited than in bacteria with low Lb content (C_Lb+_) (Figure 3).

The general formula of thiol-containing DNICs is [(RS^−^)_2_Fe^+^(NO^+^)_2_]. In Hb, from mammalian erythrocytes, one of DNIC ligands is SH-group of cysteine (cysβ93) [55,56,57,65], while another binding site has not yet been determined. The amino acid residues involved in DNICs formation in soybean Lb are also not known, since there are no available cysteine residues in this protein. Most likely, DNICs bind there to the imidazole of histidine residues, whereas the interaction of imidazole derivatives with transition metal ions in chemical and biological systems was formerly shown [66].

## 3. Discussion

Protecting organisms against reactive oxygen species commonly occurs through inducing the synthesis of antioxidant and repairing enzymes, including catalases/peroxidases [49]. In *E. coli*, an increase in the intracellular O_2_^•−^ or H_2_O_2_ concentrations is accompanied by the activation of the transcription of *soxRS* or *OxyR* genes, encoding soxRS and OxyR transcription factors, respectively [67]. The genes of enzymes protecting against oxidative and electrophilic stress: Mn-SOD, glucose-6-phosphate dehydrogenase, nitroreductase A, ferredoxin/flavodoxin reductase, iron capture regulator Fur, etc., are under the control of *soxRS* regulon [67,68]. When activated by H_2_O_2_, *OxyR* regulon induces the synthesis of 20–30 proteins, including catalase-hydroperoxidase HPI, glutathione reductase, and alkyl peroxide reductase [69]. NO, such as O_2_^•−^, enhances the transcription of soxRS regulon genes. It is assumed that activating SoxR transcription factor by NO occurs through nitrosylation of iron-sulfur centers with the subsequent formation of DNICs [70,71,72,73]. In the reactions with NO, these clusters are destroyed with DNICs formation. Moreover, due to the nitrosylation of the iron-sulfur center [4Fe-4S]^2+^ of FNR protein, *hmp* gene is activated; it encodes the synthesis of flavohemoglobin which plays a key role in protecting bacterial cells against NO [74].

The Lb interaction with peroxynitrite is similar to the mammalian Hb [75,76]. Herold and Puppo [15,17] assumed that due to a wider and more flexible heme pocket, the decay rate of the peroxynitrite-Lb complex (LbFe^III^-OONO) is an order higher than that of Mb and Hb. It is very possible that not only oxyLb but also other Lb forms (deoxyLb and metLb) effectively catalyze peroxynitrite isomerization to nitrate.

Benzylviogen is a synthetic electron donor/acceptor used in agriculture as a herbicide, and in biology, it is widely used for modeling oxidative stress. Since Bv is an active redox agent, its biological activity is associated with the electron transfer from the components of the bacterial respiratory chain to oxygen with the formation of superoxide, which is a precursor of other ROS and RNS, including hydrogen peroxide and peroxynitrite. Therefore, unlike GSNO and *t*-BOOH, Bv activates the transcription of genes of both regulons. At the same time, by activating the protective enzymatic systems, the antioxidant and antiradical input of NO metabolites (GSNO and DNICs) to *E. coli* protection against oxidative stress may be insufficient.

In our previous study, we suggested that the priority of pro- or anti-oxidant properties of Lb expressed in *E. coli* cells is dependent on its amount [77]. In cells with the reduced synthesis, Lb mostly functions as a pro-oxidant, in cells with the intensive one—as an antioxidant. It is implicitly confirmed by the data of present study, e.g., *E. coli* cells expressing Lb at low level (C_Lb+_) had a greater sensitivity to *t*-BOOH than C_Lb++_ ones (Section 2.2).

However, summing up, in bacterial cells with such effective antioxidant enzymes as superoxide dismutase and catalase/peroxidase, the antioxidant effects of Lb and DNICs may not be significant, despite the fact that DNICs and Lb are able to deactivate such oxidizers as organic hydroperoxides and peroxynitrite in cells-free systems.

Besides that, Lb is an alien protein for *E. coli*. Thus, to synthesize it, the cell consumes the energy and plastic resources required for the synthesis of de novo antioxidant enzymes, necessary under ROS and RNS action. In addition, the situation is aggravated with ferryl and oxoferrylLb, formed in reaction with peroxides. These superoxidated Lb forms can oxidize membrane lipids, which could increase oxidative stress in the cell. Therefore, such cells could have a decreased adaptive potential, especially under stress conditions.

## 4. Materials and Methods

Materials used: *Escherichia coli* TB-1 strain, modified with pEMBL18^+^::SyLb*a* plasmid with incorporated gene of soybean Lb*a* (constructed by the employees of the Autonomous University of Morelos State (Cuernavaca, Mexico), and the University of Nebraska State (Lincoln, NE, USA), and was kindly provided by Dr. Raul Arredondo-Peter from the Autonomous University of Morelos State (Cuernavaca, Mexico); Luria–Bertani (LB) medium, phenylmethylsulfonyl fluoride (PMSF)—“AppliChem” (Darmstadt, Germany); sperm whale myoglobin, DEAE-Servacel, Coomassie brilliant blue G-250—“Serva” (Heidelberg, Germany); reduced L-glutathione (GSH), 4-hydroxy-2,2,6,6-tetramethylpiperidine-1-oxyl (TEMPOL), *t*-BOOH, ferrous sulfate (FeSO_4_·7H_2_O), sodium dithionite (Na_2_O_4_S_2_), *o*-dianisidine, benzylviologene, pyridine, bovine serum albumin, human methemoglobin (metHb)—“Sigma-Aldrich” (St. Louis, MO, USA); Ultrogel AcA 54—“LKB” (Uppsala, Sweden); ampicillin—“ICN” (Beverly Hills, CA, USA). GSNO was synthesized by mixing equimolar amounts of glutathione and NaNO_2_ just before adding to the flasks. GSNO concentration in aqueous solutions was defined by optical absorption at 335 nm (molar absorption coefficient = 774 M^−1^ cm^−1^).

### 4.1. Cultivation of E. coli Cells and Treatment with Redox-Active Compounds

The study was carried out on the *E. coli* cells (TB-1 strain) with the embedded gene of soybean Lb*a*. The strain without a plasmid was used as a control. A cell suspension (50 µL) was stored in the glycerol solution at −70 °C. The cells were seeded into Luria-Bertani (LB) liquid medium containing bactotryptone (10 g/L), the yeast extract (5 g/L), NaCl (10 g/L), and ampicillin (50 mg/L in the case of modified strain). They grew for 15 h at 37 °C on a thermoshaker (“BioKom”, Moscow, Russia) with an orbital motion frequency of 200 rpm.

The obtained culture was used for the inoculation of 350 mL flasks containing 60 mL of LB liquid medium (flask filling factor 0.17) to optical density at 600 nm (OD_600_) = 0.02–0.03 in a 0.1 cm cuvette. When cultivating the Lb-synthesizing strain, the ampicillin was added.

The solutions of *t*-BOOH, Bv, and GSNO, sterilized by filtration, were added at various concentrations into the medium at the early stage of the logarithmic growth phase, 3 h after seeding (at OD_600_ of cell suspension = 0.12–0.15). The cell concentration was measured at OD_600_ of the cell suspension.

### 4.2. Spectra of the Whole Cells

The cells were washed from the components of the culture medium in 0.1 M K-phosphate buffer (pH 7.2) by the double centrifugation at 10,000 rpm for 5 min. The absorption spectra of the cells were recorded on the spectrophotometer Beckman Coulter DU 650 (Brea, CA, USA) in 1 cm quartz cuvette compared with the LB medium. The spectra were recorded at room temperature.

### 4.3. Total Protein Extraction

To obtain the protein extract, 7 h-old cell culture was used. The cell suspension (15 mL) was washed as described in Section 4.2. The cells in the washed suspension (5 mL) were destroyed with the ultrasonic disintegrator “Soniprep 150” (“MSE”, London, UK) (4 cycles for 20 s each), at the maximum power of 15 kHz. The suspension of destroyed cells was centrifuged at 15,000 rpm for 20 min. In total, 10 µL of 10 mM PMSF solution was added to the supernatant.

The supernatant was measured for its protein content using the Bradford method [78] before being used for the subsequent experiments. Here, 990 mL of H_2_O and 5 mL of the reagent were added to 10 mL of protein solution. After 5 min, the absorption was measured at 595 nm compared with the pure reagent. Protein concentration was calculated using the ratio: C (µg/mL) = A_595_ × 100/0.0042. This ratio is correct up to A_595_ = 0.4. In the case of A_595_ ≥ 0.4, the initial protein solution was 2 or 4 times diluted. The coefficients for calculating the protein concentration were determined using a standard solution of the bovine serum albumin. The reagent was prepared in the following way: of Coomassie brilliant blue G-250 (100 mg) was dissolved in 50 mL of 95% ethanol, and 100 mL of 85% phosphoric acid was added and brought to 1 L with water.

### 4.4. Determination of Lb Concentration

The Lb concentration was determined using a modified pyridinehemochromogen method [79]. The pyridine solution was prepared by mixing 100 mL of pure pyridine with 30 mL 1 M NaOH solution and was put to 300 mL volume with water. Here, 0.9 mL of pyridine solution was added to 0.3 mL of protein solution and mixed, and the total volume was divided to 2 parts. In the first part, the formed pyridinehemochromogen was reduced with sodium dithionite, and in the second one, it was oxidized with potassium ferricyanide. The optical absorption of the reduced pyridinehemochromogen was measured against the oxidized one at 556 and 539 nm. Then, the difference was calculated: ΔD = (D_556_red − D_556_oxi) − (D_539_red − D_539_oxi), and Lb concentration was defined using the mM extinction coefficient ΔE = 4.3 mM^−1^ cm^−1^, determined in our work for this ΔD [80]. Sperm whale Mb was used as a standard protein.

### 4.5. Determination of Peroxidase Activity

To assess the total peroxidase activity, a protein solution was prepared as described in Section 4.3. In total, 600 µL of *o*-dianisidine and 550 µL of water were added to 100 µL of the protein solution. The mixture was incubated for 5 min at 30 °C, 50 µL of fresh 0.075% H_2_O_2_ was added and then incubated for another 3 min, and the reaction was stopped by adding 200 µL of 50% H_2_SO_4_. The final volume of the protein solution was 1.5 mL.

The protein sediment was separated by centrifugation at 10,000 rpm for 5 min, and supernatant was used for measurement of the total peroxidase activity. It was determined spectrophotometrically by measuring the change of the color intensity of the *o*-dianisidine oxidation product at 454 nm (E_454_ = 30 mM^−1^ cm^−1^).

The *o*-dianisidine solution was prepared as follows: 200 mL of 1% alcohol solution of *o*-dianisidine was added to 5 mL of 0.4 M K-phosphate buffer (pH 5.9), and the mixture was diluted to 20 mL with water. The reagent was stored without access to light for ≤2 weeks.

### 4.6. Isolation and Purification of Lb from E. coli Cells

To obtain the Lb-containing fraction, a cell culture (15 h growth), pre-washed of the medium components, was destroyed with the ultrasonic disintegrator “Soniprep 150” (“MSE”, London, UK) at the maximum power of 23 kHz. The solution for destruction included 50 mM Tris-HCl buffer (pH 7.4), lysozyme (10 mg/mL) and 20 µM PMSF. The proteins of the cell extract were precipitated with ammonium sulfate in 50–90% saturation range. Lb*a* was purified in 2 stages. The first was performed on the Ultrogel AcA-54 column (1 × 60 cm) with 0.02 M K-phosphate buffer (pH 7.2). To inhibit the protease activity PMSF was added to the final concentration 10 µM. Further Lb purification was carried out on DEAE cellulose column (7 × 1 cm) in 0.01 M K-phosphate buffer (pH 7.2), with the linear NaCl gradient from 0 to 1 M. The Lb*a* fraction was eluted at 0.3 M NaCl, concentrated by ultrafiltration, oxidized with potassium ferricyanide, and dialyzed against 0.02 M K-phosphate buffer (pH 7.2). All procedures were carried out at 4 °C. If necessary, the protein was concentrated on Millipore membrane in ultramembrane concentrating cell “Amicon” Model 12 (“Amicon”, Walnut Creek, CA, USA) under argon gas flow.

### 4.7. Measurement of Lb Peroxidase Activity in Relation to t-BOOH

Peroxidase activity of recombinant Lb was measured by color intensity of *o*-dianisidine at 454 nm (ε_454_ = 30 mM^−1^ cm^−1^), oxidized with *t*-BOOH. The reaction mixture contained 0.1 M K-phosphate buffer (pH 7.4), 8 µM Lb, 0.5 mM *t*-BOOH, and 0.8 mM *o*-dianisidine. The measurements were carried out at 25 °C on “Cary” spectrophotometer (“VarianBio”, Palo Alto, CA, USA).

### 4.8. Synthesis of Dinitrosyl Iron Complexes

Paramagnetic DNICs with phosphate ligands were obtained by treating 5 mM iron sulfate in 100 mM K-phosphate buffer (pH 6.8) by gaseous NO in Thunberg vessel, containing 100 mL gas phase. In total, 1 mL of FeSO_4_ solution in distilled water (pH 5.5) and 4.5 mL of phosphate buffer were placed into lower and upper parts of the Thunberg vessel, respectively. The vessel was sealed, and gaseous NO was added to a partial pressure of 100 mm Hg. Then, the solution was mixed for 5 min, and NO was removed from the Thunberg vessel.

Lb-DNICs were prepared by adding 200 mL of 5 mM phosphate DNICs to 500 mL of 0.5 mM Lb solution in 0.1 M K-phosphate buffer (pH 7.2). Lb-DNICs were formed after 5 min incubation.

### 4.9. Registration of DNICs by EPR Method

DNICs in the cells and in the model systems with purified Lb and Hb were registered using the EPR spectra recorded at room temperature (~20 °C) or at −170 °C on E-109E X-band spectrometer (“Varian”, Palo Alto, CA, USA). Samples (80 µL) were placed in glass capillaries. The measurement parameters were as follows: modulation frequency, 100 kHz, microwave radiation power, 10 mV, microwave radiation frequency, 9.15 GHz and modulation amplitude, and 0.2 and 0.1 mT, for the spin adducts of DNICs and DEPMPO, respectively.

The concentration of Lb-DNICs was estimated by EPR signal of the complexes, using TEMPOL spin label as a standard.

### 4.10. Registration of Dinitrosyl Iron Complexes in Whole Cells

To apply EPR spectra, *E. coli* cells were grown for 3 h at the standard conditions, 6 times concentrated by centrifugation and resuspended in 0.1 M phosphate buffer (pH 7.2). The cell suspension was treated with GSNO solution (final concentration 0.5 mM) and was incubated for 20 min at 37 °C. To induce the oxidative stress, *t*-BOOH was added to a number of flasks 5 min after GSNO to a final concentration 23 mM. Then, the cells were precipitated by centrifugation and resuspended in small volumes of the same buffer. The concentration of cells in the final suspension was 75 times higher than of initial culture. The 0.8 mL aliquots of concentrated cell suspension were directly placed to plastic EPR tubes and immediately frozen in liquid nitrogen (−196 °C). EPR spectra were recorded at −170 °C. The spectra of DNICs with Hb and Lb were also recorded at room temperature (20 °C). The amount of DNICs was estimated by the intensity of characteristic EPR signal of paramagnetic DNIC form (g = 2.03). EPR spectra were recorded by E-109E spectrometer (“Varian”, Palo Alto, CA, USA) at the following conditions: microwave power, 10 mW; microwave frequency, 9.30 GHz; modulation amplitude, 0.4 mT; and receiver gain, 10^5^.

All experiments were performed in no less than 3 replicates. The results are presented as an average ± standard deviation.

## 5. Conclusions

This work was devoted to the study of the effect of the Lb expression on the susceptibility of *E. coli* cells to oxidative and nitrosative stress (redox stress). Lb is a member of the family of hemoglobins—proteins, performed a wide range of functions: storage and transport of oxygen and NO, detoxification of organic and inorganic peroxides and peroxynitrite, regulation and detection of NO concentration, and other enzymatic activities. However, these proteins can also pose a threat to the cell, since their prosthetic heme group contains an iron atom that can initiate the formation of free radical metabolites (ROS and RNS), damaging the protein itself and other biomolecules. Whether Hb acts as an antioxidant or a pro-oxidant depends on the functional state and concentration of Hb and on redox conditions. In the bacterial system used, the pro-oxidant effects of Lb prevailed over the antioxidant ones. The results obtained can help for understanding the mechanisms of Lb functioning both in nodules (legume-rhizobial nitrogen-fixing symbiosis) under environmental stress and in the microorganisms used for synthesizing Lb and other hemoglobins for biotechnological purposes.

## Figures and Tables

**Figure 1 molecules-26-07207-f001:**
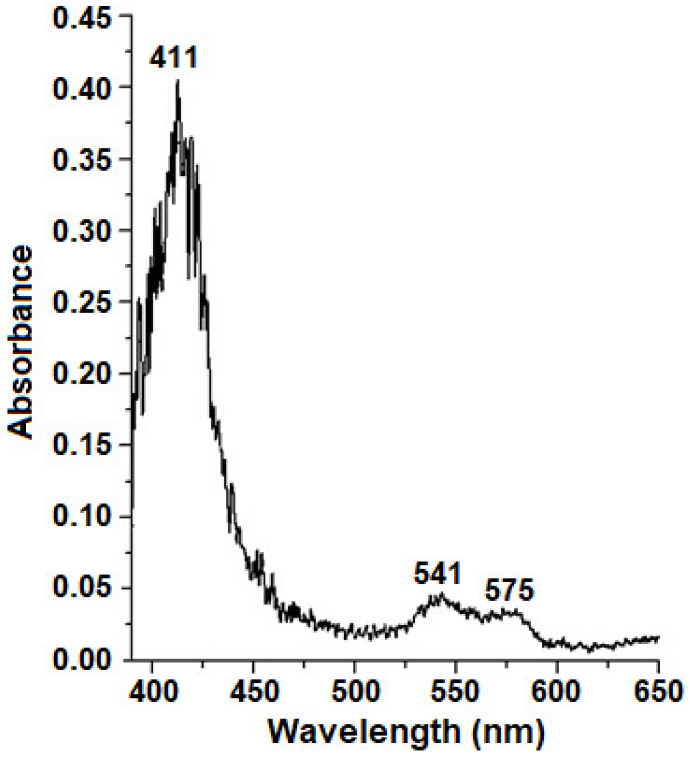
The absorption spectra of *E. coli* cells washed from medium, representing the difference between spectra of actively synthesizing Lb*a* cells (C_Lb++_) and nonsynthesizing Lb ones (C_Lb−_): the spectrum of C_Lb++_ cells minus the spectrum of C_Lb−_ cells.

**Figure 2 molecules-26-07207-f002:**
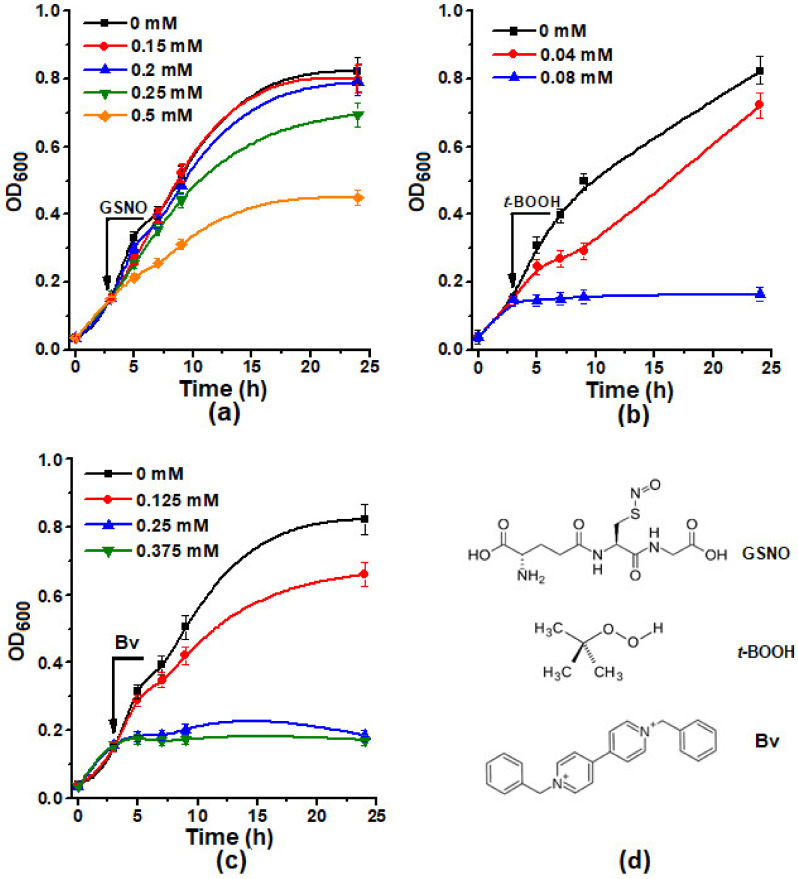
Growth curves of *E. coli* (C_Lb++_) at different concentrations of S-nitrosoglutathione (GSNO) (**a**), *tert*-butyl hydroperoxide (*t*-BOOH) (**b**), and benzylviologene (Bv) (**c**). Inducers of oxidative and nitrosative stress were added after 3 h of *E. coli* growth. Structural formulas of GSNO, *t*-BOOH, and Bv are shown on (**d**).

**Figure 3 molecules-26-07207-f003:**
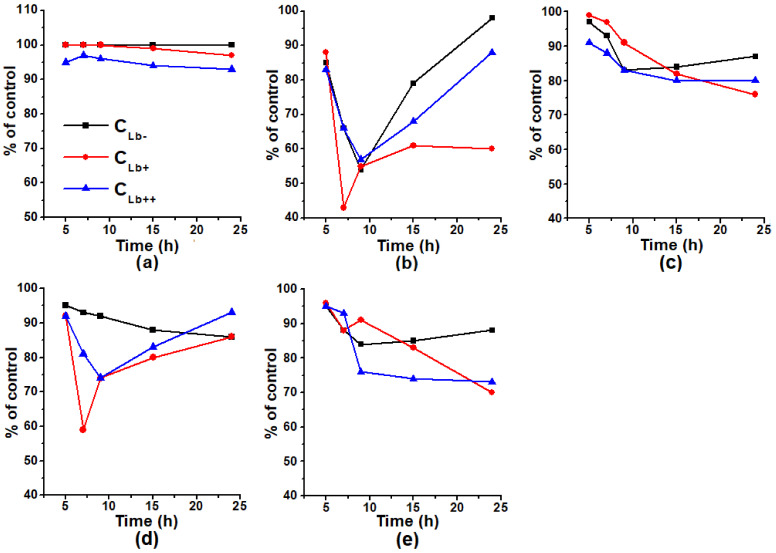
The growth of the bacterial culture, in percentage of the control culture growth (without adding inducers of oxidative and nitrosative stress). Inducers of oxidative and nitrosative stress were added after 3 h since the flasks sowing in concentrations: GSNO—0.2 mM, *t*-BOOH—0.04 mM, Bv—0.125 mM. Picture panels: GSNO—(**a**), *t*-BOOH—(**b**), Bv—(**c**), GSNO + *t*-BOOH—(**d**), and GSNO + Bv—(**e**).

**Figure 4 molecules-26-07207-f004:**
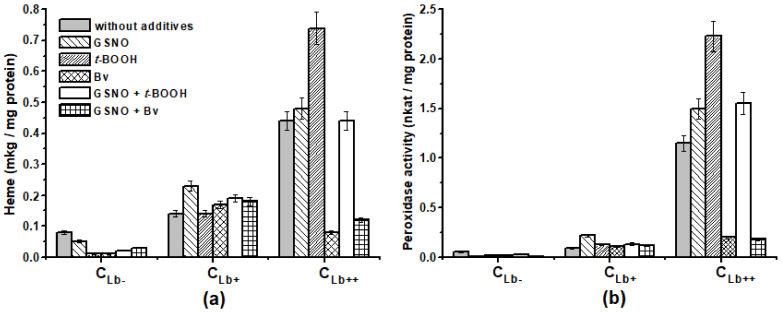
The concentration of heme—(**a**), and the total peroxidase activity—(**b**) of the cell lysate proteins in the middle of the logarithmic growth phase (4 h after the addition of the inducers of oxidative and nitrosative stress—7 h of growth).

**Figure 5 molecules-26-07207-f005:**
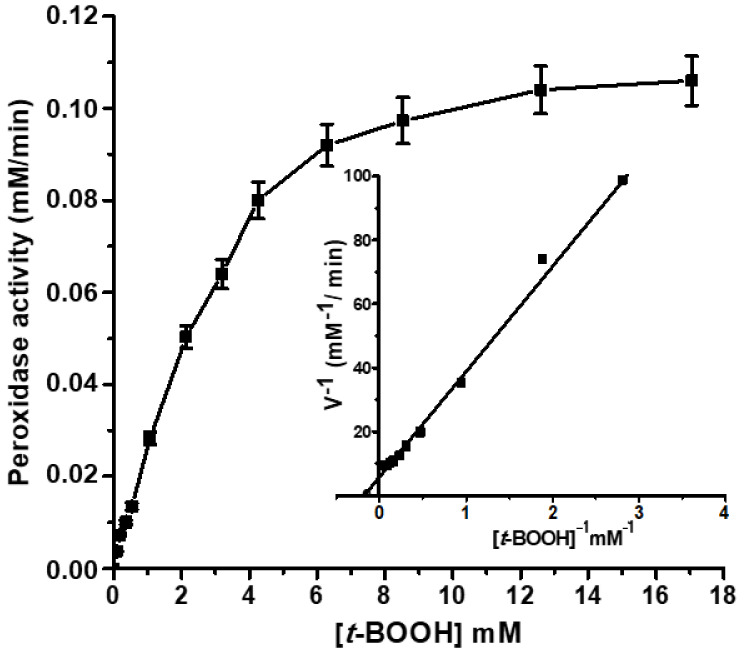
The dependence of the leghemoglobin (Lb) peroxidase reaction rate on the concentration of the substrate (*t*-BOOH). The insert shows the dependence in the Linuiver–Burke coordinates. Peroxidase activity was measured in the presence of 8 µM Lb and 0.8 mM *o*-dianisidine.

**Figure 6 molecules-26-07207-f006:**
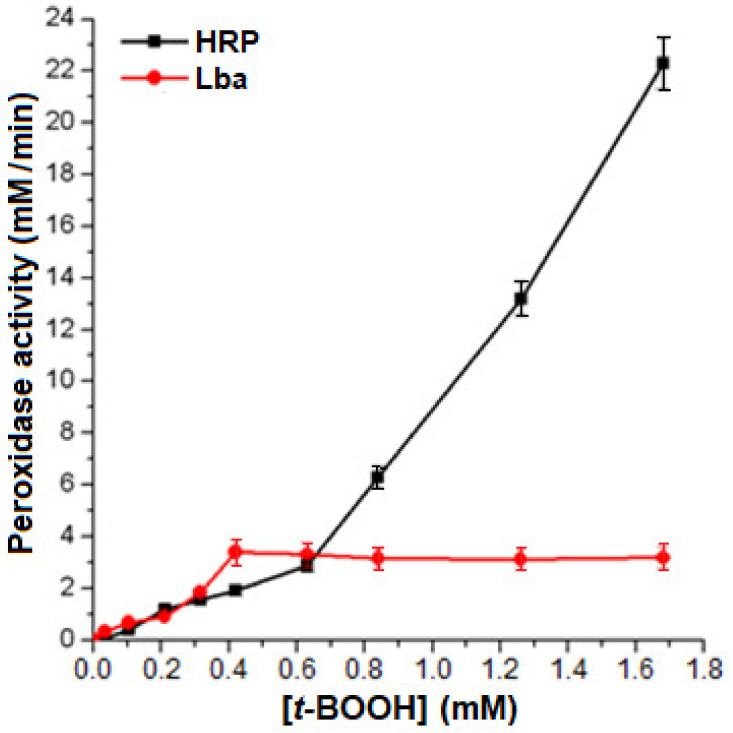
The dependence of initial rate of the peroxidase reaction catalyzed by Lb*a* and by horseradish root peroxidase (HRP) on the concentration of *t*-BOOH. Peroxidase activity was measured in the presence of 8 µM Lb or 0.8 µM HRP and 0.8 mM *o*-dianisidine.

**Figure 7 molecules-26-07207-f007:**
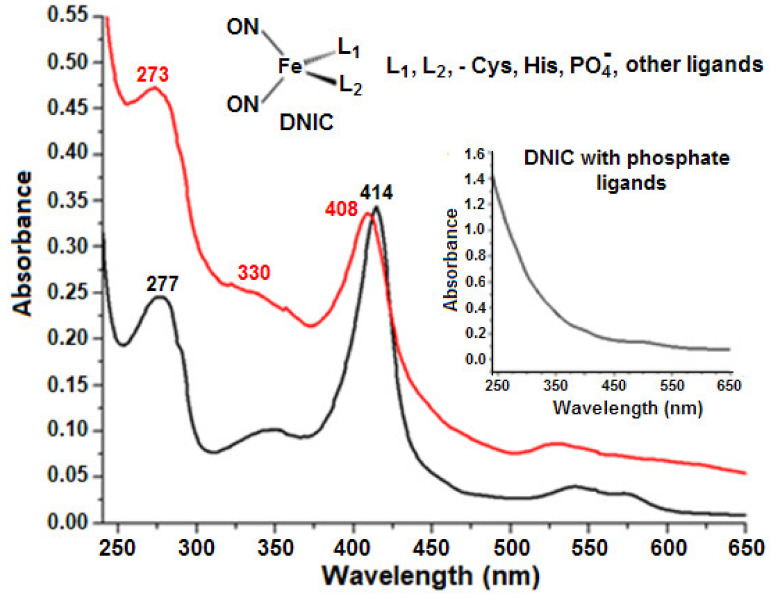
Absorption spectra of recombinant Lb*a*—the black curve—and its complex with DNIC—the red curve. The insert shows the spectrum of dinitrosyl iron complexes (DNICs) with phosphate ligands.

**Figure 8 molecules-26-07207-f008:**
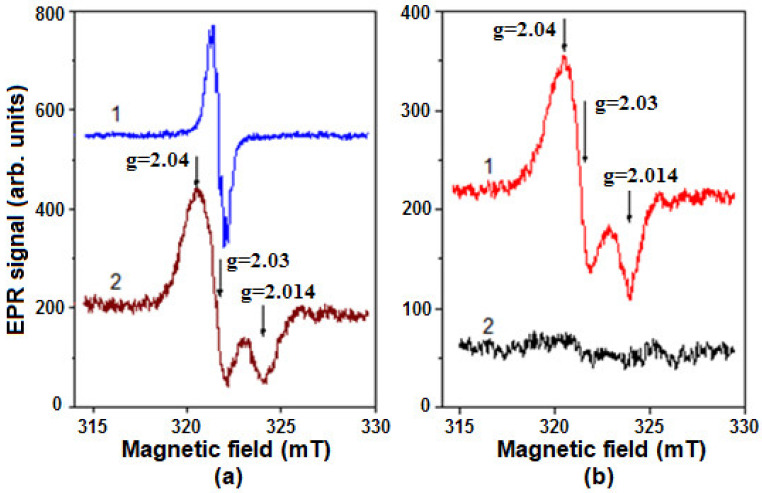
Electron paramagnetic resonance (EPR) spectra of DNICs bound with human methemoglobin (metHb) and soybean metLb. (**a**): 1—the spectrum of DNICs with phosphate ligands used to produce protein DNIC; 2—spectrum of Hb-bound DNICs (Hb-DNICs). (**b**): 1—spectrum of Lb-bound DNICs (Lb-DNIC); 2—the same as (1) + 1 mM *t*-BOOH. Spectra recorded at room temperature.

**Figure 9 molecules-26-07207-f009:**
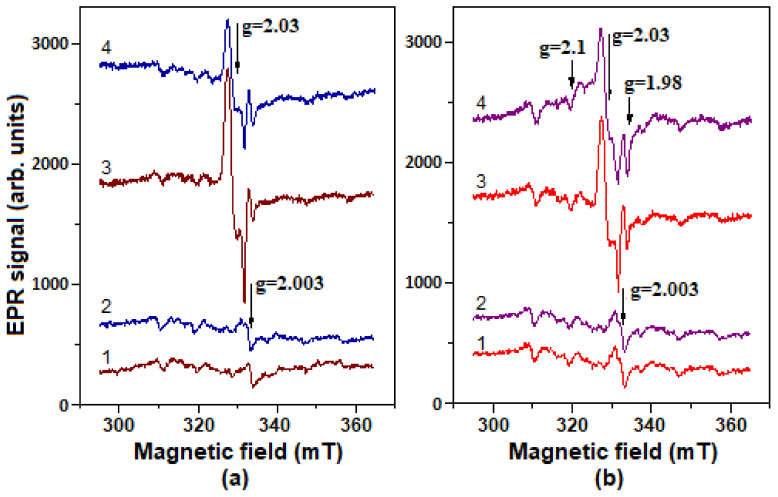
Low-temperature EPR spectra of *E. coli* cells: C_Lb−_ (**a**) and C_Lb++_ (**b**). 1—untreated bacteria; 2—bacteria after incubation with *t*-BOOH; 3—after incubation with GSNO; 4—after incubation with GSNO + *t*-BOOH. The registration was carried out at −170 °C.

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
