# Peer review of "Expressed Soybean Leghemoglobin: Effect on *Escherichia coli* at Oxidative and Nitrosative Stress"

_molecules, 2021, doi:10.3390/molecules26237207_

Round 1
Reviewer 1 Report
The article expressed soybean leghemoglobin: effect on Escherichia coli at oxidative and nitrosative stress is a well-written article, with somehow one or two rephrasing needs, for example at the beginning of page 4, O-O bond does not only beaks, if yes, it would afford OH. and not OH-. However, the article affords hypotheses largely confirmed by experience. For publication in the journal "molecules" I would suggest for readers ease that some molecular drawing, especially Lb, Hb, GSNO, benzylviologen etc... would be inserted.
Overall I recommend this article for publication.
Author Response
We want to thank eminent reviewer for hard job on reviewing our manuscript and for valuable comments which would help us to improve it.
We hope that changes made will meet the reviewer's requirements.
Please see the attachment.

Reviewer 2 Report
The manuscript aimed to investigate the interaction of leghemoglobin with stress inducers in lb-expressing E. coli. The work appears to have been performed in an acceptable manner. However, it will be useful to the readers if the following points are considered and addressed.
Major concerns:
Moderate linguistic (English) corrections are desirable. The submitted version has numerous missing articles and sentence framing/ phrase construction issues.
Was the plasmid construction been carried out in this study, or it was from the previous work? Where or how did the authors obtain the soybean Lba? Has the sequence been deposited in the gene bank (with accession number)?
Overall, the authors have provided detailed information about leghemoglobin and its interaction/reaction with inducers. However, some information can be scattered, and in my opinion, it could be more organized. I appreciate the way of presenting a brief introduction in the Results section before reporting the findings. But I feel that some are too lengthy, and it seems that this "introduction" is more suitable to be placed under Introduction or Discussion sections. The discussion might be too general. Probably, the authors could try to relate it to the findings in this study.
Minor concerns:
Abstract
L12: Please include the main aim/goal of the study.
L17: "also pinpointed" change to "detected"
L20: "We suppose that Lb synthesis as an..." change to "This might be due to the energy demand required by the Lb synthesis"
L21: how does lb contribution to ROS metabolism could aggravate the situation? This sentence seems unclear without reading the results and discussion in this manuscript.
Introduction
L31: "form and properties" Is this referring to Lb? Please include.
L32: "O2" Please define it for the first time (although it is a standard molecular formula). The authors could use this formula in the subsequent text, e.g. L34, 35, etc. The same goes for other molecular formulas, e.g. L50 NO (nitric oxide), L164 and others.
L45: spacing issue, between "also" and "in". Also in L97
L53: "reactive oxygen (ROS) species" change to "reactive oxygen species (ROS)"
L66: "faster, releasing ion, which, in its turn.." Faster compared to? Please rephrase.
L88-90: This point has been discussed in the Discussion section.
Results
L102: "physiologically active state" Is the term "physiologically" appropriate? How to determine it physiologically? Or just put "active state"?
L105: Figure 1: Please define Clb++ and Clb- in the caption.
L106-117: This paragraph seems like an introduction or discussion.
L121: "to maintain" Would it be better to use "to induce"? If to maintain, which means the authors induced oxidative stress to the cells and to maintain it, the authors have added these substances again after 3 hours.
L148: "additives" change to "adding"
L154 and 166: Should it be Figure 3c or 3d?
L167: “even at t-BOOH concentrations” Concentrations? Here only has one concentration. Please rephrase.
L186: I am afraid I may not understand this correctly. It seems that GSNO does not affect the reaction of Bv and GSNO. However, if the GSNO can react with the produced peroxynitrite, wouldn't it also has some effects (even though peroxynitrite is a strong oxidizer)? Perhaps the authors could clarify this.
L198: Is this referring to Figure 4b? If yes, I suggest including "Figure 4b".
L201: Figure 4a or 4b?
L202: "several times" Be specific.
L207: "Those changes" refers to which changes? A new paragraph should not start with this word. If this is referring to Bv and GSNO reactions, I suggest combining these two paragraphs.
L211: Figure 4, Is there any significant difference for each type of cell culture when compared to untreated cells? Do you think it would be useful to analyze it statistically?
L323: Figures 8 and 9, I wonder why the CLb+ was not included in these experiments?
Discussion
L404: "highly successful" in terms of?
Perhaps the authors could consider including a description of the impact/significance of the study in the last paragraph? How could these findings help the community or fill out the knowledge gap?
Methods
L432: "for seeding a series of 350 ml flasks" Please rephrase.
L439: "by" change to "at"
L448 and 478: Which paragraph 2? I suggest stating which sections, e.g. 4.2?
L446 and 454: I think these two sections can be combined. "0.3 ml of the supernatant was taken for measuring total peroxidase activity" could be replaced with "The supernatant was measured for its protein content using Bradford method [78] before being used for the subsequent experiments". The subheading for section 4.3 can change to "Total protein extraction".
L477: Please specify the volume for the protein solution, i.e., 0.3 ml.
L483: "by the color intensity" change to "by measuring the change of the color intensity"
L495: "1*60 cm" Not "1 × 60 cm"?
L496: "PMSF was added to 10 µM concentration" A bit unclear. Please consider rephrasing.
L507: "t-BOOH" t should be in italic form.
L522: What is the model system? Untreated? If yes, I suggest changing it to untreated. Same to L82.
L537: What is *75??
Author Response
We thank eminent reviewer for hard job on reviewing our manuscript and for valuable comments which would help us to improve its quality.
We hope that changes made will meet the reviewer's requirements.
Please see the attachment.
